# On-chip petahertz electronics for single-shot phase detection

Felix Ritzkowsky [1,2,6] ✉, Matthew Yeung[2,6], Engjell Bebeti[1], Thomas Gebert[3,4], Toru Matsuyama[3], Matthias Budden [3,4], Roland E. Mainz [1], Huseyin Cankaya[1], Karl K. Berggren [2], Giulio Maria Rossi[1], Phillip D. Keathley [2] ✉ & Franz X. Kärtner [1,5]

Attosecond science has demonstrated that electrons can be controlled on the sub-cycle time scale of an optical waveform, paving the way towards optical frequency electronics. However, these experiments historically relied on high-energy laser pulses and detection not suitable for microelectronic integration. For practical optical frequency electronics, a system suitable for integration and capable of generating detectable signals with low pulse energies is needed. While current from plasmonic nanoantenna emitters can be driven at optical frequencies, low charge yields have been a significant limitation. In this work we demonstrate that large-scale electrically connected plasmonic nanoantenna networks, when driven in concert, enable charge yields sufficient for single-shot carrier-envelope phase detection at repetition rates exceeding tens of kilohertz. We not only show that limitations in single-shot CEP detection techniques can be overcome, but also demonstrate a flexible approach to optical frequency electronics in general, enabling future applications such as high sensitivity petahertz-bandwidth electric field sampling or logic-circuits.

When John A. Fleming developed the first widely usable vacuum diode based on the thermionic emission of electrons from a tungsten filament and showed for the first time the rectification of electronic AC signals, he laid the foundation for modern electronics[1]. Around one hundred years later, in the pursuit of ever faster electronics, a major advancement was made by utilizing carrier-envelope phase (CEP) controlled few-cycle pulses to rectify electric fields at hundreds of terahertz at sharp metal tips[2]. This not only demonstrated the generation of rectified, optical-frequency currents but also demonstrated control over attosecond electron currents by controlling the optical pulse CEP. Subsequent investigations into these emission processes revealed complex attosecond-fast dynamics[3,4].

With the goal of achieving electronics operating at the frequency of optical waves, many methods have been investigated for generating rectified femto-to-attosecond currents directly in closed electric circuit elements. For example, by using sub-cycle interband transitions in dielectrics[5–8], or metallic nanoantennas[9,10]. These steps toward integrated circuits significantly reduced the experimental requirements from large and bulky vacuum equipment to low-energy ambient operation. Applications exploiting the sub-cycle nature of these currents have been demonstrated. Examples include attosecond-resolution electric field measurements, CEP detection of few-cycle pulses, and petahertz logic gates[6,8–17]. Specifically, CEP detection presents a great testbed for petahertz electronics, as previous methods have been fundamentally limited, such as f-2f-interferometry lacking sensitivity for the absolute CEP or gas-ionization-based methods, that do provide absolute CEP sensitivity but require microjoule level pulses. Resonant nanoantennas have emerged as an attractive option, as they significantly reduce the energy required for field emission by optical pulses and present a physical reference for the absolute CEP[9,10,12,18–20].

[1]Center for Free-Electron Laser Science CFEL, Deutsches Elektronen-Synchrotron DESY, Hamburg, Germany. [2]Research Laboratory of Electronics, Massachusetts Institute of Technology, Cambridge, MA, USA. [3]Max Planck Institute for the Structure and Dynamics of Matter, Hamburg, Germany. [4]WiredSense GmbH, Hamburg, Germany. [5]Department of Physics and The Hamburg Centre for Ultrafast Imaging, Universität Hamburg, Hamburg, Germany. [6]These authors contributed equally: Felix Ritzkowsky, Matthew Yeung. ✉e-mail: felix.ritzkowsky@desy.de; pdkeat2@mit.edu

This reduction can reach up to three orders of magnitude, lowering the energy requirement to picojoule levels, while confining electron emission to a well-defined hotspot at the sharp tip of the nanoantenna. In addition, by exploiting the extreme spatial confinement of nanoantennas, attosecond time-scale charge transport across nanometer-sized junctions has been achieved[21].

While resonant nanoantennas offer several advantages, they also have limitations that impact their practicality. To the best of our knowledge, the electron yield from these nanoantennas has never exceeded one electron per shot in CEP-sensitive yield[2,9,10,12,18]. As a result, thousands of individual laser shots must be integrated to achieve a statistically significant signal, which means high-repetition-rate laser sources are required. Ideally, enough current would be generated per laser shot for CEP-sensitive readout without the need for averaging. Simply increasing the peak intensity of the laser pulse cannot scale the signal level of these devices, as this would cause irreversible laser-induced damage. To circumvent damage, the pulse energy can be distributed over a network of nanoantennas, which respond individually at a PHz bandwidth to the optical field but collectively contribute their produced charge signal to the network, which is subsequently read out at radio frequencies. However, scaling up the number of nanoantennas in a single network has been shown to present difficulties as fabrication variance couples to the detected CEP signal and reduces the overall signal strength[18]. Second, large variations of intensity across the network might exhibit CEP vanishing points that either cause a vanishing CEP signal when the local intensity hits a waveform-specific resonant intensity or even cause a $\pi$ phase shift for intensities above that resonance[22].

In this work, we overcome these issues and demonstrate single-shot detection of CEP-dependent electrons generated by optical tunneling in a fully on-chip nanoantenna device for shot-to-shot carrier-envelope phase detection. We achieve this through the simultaneous excitation of hundreds of interconnected off-resonant metallic nanoantennas[18]. This approach enables coherently driven, attosecond-timescale electron emission across the entire detector area of 225 $\mu m^2$. Moreover, by employing a custom-developed mid-infrared (MIR) sub-2-cycle laser source[23], we obtain a more than tenfold increase in charge

emission per individual antenna compared to previous results, with a CEP-sensitive charge emission as high as 3.3 electrons per shot per antenna[18]. Optical pulses with longer central wavelengths have a proportionally higher electron yield per individual half-cycle compared to their shorter-wavelength counterparts. In addition, the longer wavelength driver excites the nanoantenna off-resonantly, which enables the full reproduction of the incident electric field at the nanoantenna tip. The off-resonant excitation is crucial, as the number of optical cycles dramatically influences the amount of CEP-sensitive charge produced[18]. Through this combination of short-pulse excitation and scaling of the emitter area, we achieve, to the best of our knowledge, the highest ever recorded CEP-sensitive charge yield from an integrated petahertz electronic detector and a single laser shot, achieving in excess of 2300 e per laser shot at the full repetition rate of the laser system (50 kHz). The energy requirements of less than 100 nJ represent a reduction of 2 to 3 orders of magnitude compared to alternative gas-phase methods while removing the need for vacuum conditions[15,24]. Such devices enable compact, shot-to-shot CEP detection for various attosecond experiments that require CEP diagnostics[25–27]. Our work more broadly demonstrates the viability of low-energy, chip-scale petahertz-electronics with single-shot readout.

## Results

### Device design

Our devices, as seen in Fig. 1a, consist of 722 interconnected metallic (Au) bow-tie nanoantennas embedded in a 15 $\mu m$ by 15 $\mu m$ network. The device is integrated into an off-chip readout circuit using conventional electronics. The individual bow-tie nanoantennas, as shown in the scanning electron microscope image in Fig. 1c, have designed dimensions of 530 nm in length, 142 nm in width, and 20 nm in thickness, resulting in an antenna density of 3.2 $\mu m^{-2}$. Figure 1d shows the finite element electromagnetic simulation of the field distribution, showing a peak enhancement of up to ~18-fold for 111 THz (2.7 $\mu m$ wavelength) localized at the tips of the bow-tie structure. The sharp antenna tip creates a spatially confined hot spot for electron emission to occur. When the whole network is illuminated with a few-cycle infrared laser pulse with a peak electric field on the order of 1 V nm$^{-1}$,

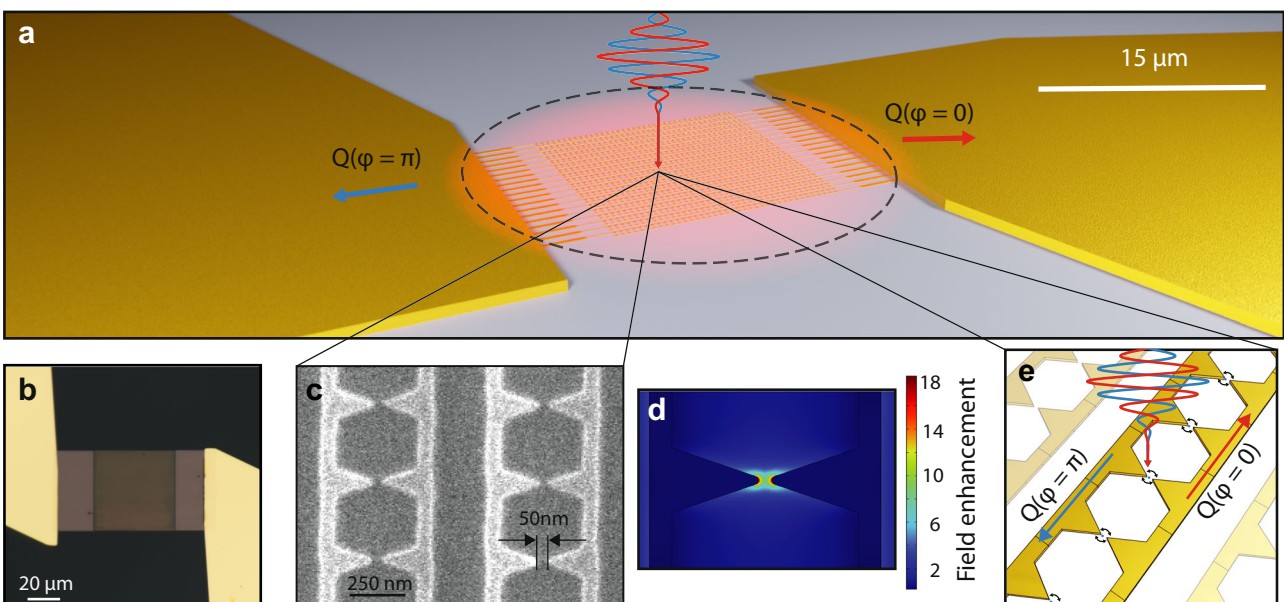

**Fig. 1 | CEP-dependent charge generation in nanoantenna arrays. a** Schematic of the charge generation process in the network showing two electric fields with a $\pi$ CEP shift corresponding to charge generated with positive $Q(\varphi = 0)$ or negative sign $Q(\varphi = \pi)$. **b** Optical microscope image of an integrated nanoantenna network contacted with gold leads. **c** Scanning electron microscope image of a metallic

nanoantenna array. **d** Finite-element method simulation using COMSOL of the spatial field enhancement distribution of a single antenna pair. **e** Schematic of the nanoscopic emission process, showing the sub-cycle electron currents generated in the antenna-vacuum junction by the driving field.

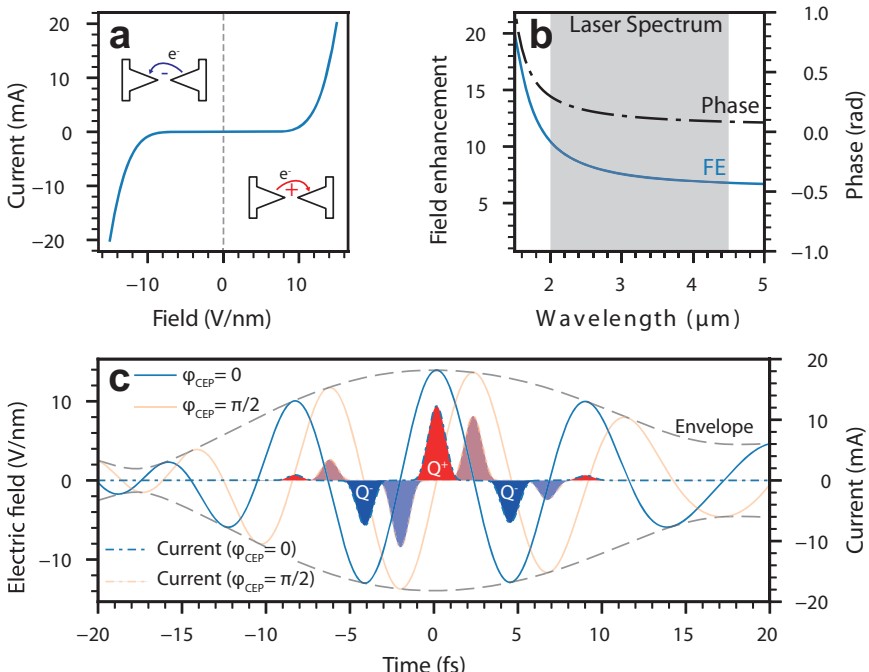

**Fig. 2 | Theoretical description of the antenna gap currents. a** Effective instantaneous tunneling rate for two opposing gold surfaces in the nanoantenna junction, assuming scaling parameters from[33] with an effective emission area of 628 nm². **b** The response function of the local electric field at the tip of the nanoantenna to an exciting electric field simulated using a FEM electromagnetic solver. The simulation shows the wavelength-dependent field enhancement and phase. The effective field enhancement of the incident pulse is ~8.2. **c** The electric field as a function of time and the calculated instantaneous current as a function of the electric field for a CEP of $\varphi = 0, \pi/2$. The electric field is the calculated local antenna field using the characterized optical pulse and the simulated antenna response (see Electromagnetic simulation of the nanoantenna and Experimental setup in Supplementary Information). The solid lines note the electric field, and the dashed lines the current. The shaded areas underneath the current curves show the total charge yield, with red areas contributing positively and blue areas contributing negatively.

highly nonlinear tunnel ionization of electrons occurs at these hotspots at the tip of the bow-tie antennas. In addition, theoretical models predict that the tunnel ionization is temporally confined to the peak regions of the strongest half-cycles of the exciting field[3,4,12,21,22,28–30].

**Theoretical Model**

In the case of sufficiently strong electric fields, with a Keldysh parameter $\gamma \ll 1$, the tunneling emission for a metal-vacuum boundary is described by the quasi-static Fowler-Nordheim tunneling current $\Gamma_{FN}(E) = \theta(E)\alpha E^2 \exp(-\frac{f_t}{|E|})$[28–31], with $\theta(E)$ noting the Heaviside function, $f_t = 78.7$ V nm⁻¹ the characteristic tunneling field strength for gold and $\alpha$ a material and geometry dependent scaling factor. Since a single bow-tie is, in fact, a symmetric system consisting of two metal surfaces facing each other with a 50 nm vacuum gap, we can approximate the total instantaneous currents at the junction with $\Gamma(E) = \Gamma_{FN}(E) - \Gamma_{FN}(-E)$, as experimentally shown in refs. 9,18,21. A CW laser would lead to fully symmetric charge injection and transport across the gap. In such a case, the time average of the residual charge in the network is zero. However, for the case of a few- to single-cycle pulse, the highly nonlinear dependence of the tunneling current with respect to the electric field amplitude does result in a residual net charge. The residual net charge is caused by the significant amplitude differences between the individual half-cycles of the pulse, effectively breaking the symmetry of emission and transport[9]. To understand the symmetry breaking, it is useful to look at the detailed instantaneous tunneling rates as a function of the electric fields for a metal-vacuum boundary.

The instantaneous current response of this nanoantenna configuration, shown in Fig. 2a, is equivalent to the response of two parallel diodes in opposing directions. The quantitative current response is adapted from refs. 32,33 and considers the frequency-resolved field enhancement, while also averaging over the antenna tip surface area of 628 nm², resulting in an effective field enhancement of 8.2 for the considered excitation field. When calculating the instantaneous currents of the nanoantenna, the local field at the tip of the nanoantenna is relevant. Therefore, we need to consider the antenna's complex transfer function[12]. The antenna is designed to have a resonance wavelength of 1500 nm and be off-resonant with the exciting field centered at 2.7 μm for two main reasons; the first is to transfer the full bandwidth of the optical pulse to the antenna tip, as a sharp resonance would increase the local pulse duration and reduce the CEP-dependent charge yield drastically. The second reason is that the fabrication process is not fully uniform throughout the detector area, resulting in small spectral shifts of the antenna resonance[18]. When designed on-resonance, small variations will result in large phase differences between individual antennas, as the phase response has a steep slope at resonance. Considering the collective phase response of all antennas, variations in individual phases will reduce the collective CEP response of the detector[18]. Therefore, when the antennas are driven off-resonance, small variations in the fabrication will not translate into large phase changes of the optical field at the antenna tip. In addition, a reduced variance of the device-induced phase shift is critical to improved precision in measuring the absolute CEP value. Any well-known phase offset induced by the antenna can simply be removed from the detected phase. The local field enhancement and the phase response of an off-resonance antenna for wavelengths above 2 μm are shown in Fig. 2b. The local field at the antenna tip $E_{loc}$ is, therefore, the frequency domain multiplication of the incident pulse $\tilde{E}(\omega)$ and the antenna complex frequency response $\tilde{H}(\omega)$, $E_{loc}(t) = \mathcal{F}^{-1}\{\tilde{E}(\omega) \cdot \tilde{H}(\omega)\}$. The calculated instantaneous current response of the system to such a pulse with a peak field of ~13 V nm⁻¹ is shown in Fig. 2c. The employed optical pulse shape is the reconstructed optical pulse used in the experimental apparatus, combined with the simulated local field enhancement (see Electromagnetic simulation of the nanoantenna and

Experimental setup in Supplementary Information for details). This implies that the central half-cycle with the highest field amplitudes generates the largest peak current with up to 12 mA for a duration of 1.1 fs (FWHM). The neighboring half-cycles generate substantially smaller currents with the opposite sign. Since conventional electronics do not support the petahertz bandwidth currents, the device acts as an integrator, and the net charge deposited by the optical pulse resides in the circuit network, similar to a photodiode. The mathematical description of these charges $Q$ as a function of the pulse CEP $\varphi$ is simply the integral over the instantaneous currents;

$$Q(\varphi) = \int_{-\infty}^{\infty} \Gamma(A(t) \cdot \cos(\omega_0 t + \varphi)) \, dt \qquad (1)$$

$$= \underbrace{\int_{-\infty}^{\infty} \Gamma_{FN}(A(t) \cdot \cos(\omega_0 t + \varphi)) \, dt}_{:= Q^+(\varphi)} - \underbrace{\int_{-\infty}^{\infty} \Gamma_{FN}(-A(t) \cdot \cos(\omega_0 t + \varphi)) \, dt}_{:= Q^-(\varphi)}$$

$$(2)$$

$$= Q^+(\varphi) - Q^-(\varphi). \qquad (3)$$

The CEP dependence of the charge now stems from the small difference of $Q^+(\varphi)$ and $Q^-(\varphi)$. For the case of a cosine pulse ($\varphi = 0$), the charge yield becomes maximal, and for the case of a sine pulse ($\varphi = \pi/2$), the charge components cancel out to zero. Based on the results shown in ref. 18 with 0.1 e per antenna, one can anticipate CEP-dependent charge amplitudes of around 1.4 e per antenna for the optical pulses used in our experiments and a peak field of 1.7 V nm⁻¹. The resulting charge increase is due to a reduced number of cycles (from 2.5 to 2), and the use of a longer central wavelength[33]. With the known charge yield per antenna, one can extrapolate the charge yield of an array of interconnected antennas to a charge that is within the reach of reasonable detection limits.

## Experiment

The optical pulses used in this work were generated with a home-built laser source based on optical parametric amplification and difference frequency generation that delivers passively CEP stable pulses with an FWHM duration down to 16 fs at a center wavelength of 2.7 µm. The pulse energy was >84 nJ at a repetition rate of 50 kHz. The CEP of the laser was controlled by adjusting the pump-seed delay in the difference frequency generation stage. The delay adjustment was implemented by controlling the pump beam path length via a retro-reflector mounted on a piezo-actuated linear stage. For a detailed description of the source, see the "Methods" section A and ref. 23.

To illuminate the nanoantenna network, we focused the incident pulse down to ~21 µm (FWHM) with an off-axis parabola of focal length 25.4 mm (see Fig. 3). The nanoantenna arrays were placed in the center of the focus. To achieve single-shot charge readout, we used a custom trans-impedance amplifier with a gain of 1 GV A⁻¹ and a −3 dB bandwidth of 50 kHz (WiredSense GmbH). The RMS noise floor of our detection was measured to be ~1100 e per shot. To overcome this noise, we illuminated a network consisting of 722 antennas in a rectangular area of 15 µm by 15 µm to generate in excess of 1000 e per shot.

After interaction with the nanoantenna arrays, pulse energies were measured by a pyroelectric photodetector with the same −3 dB bandwidth of 50 kHz as the trans-impedance amplifier. This arrangement allowed for the simultaneous recording of shot-to-shot pulse energy fluctuations. The pyroelectric detector uses an identical trans-impedance amplifier to the one used for the nanoantenna read-out to ensure comparable statistics of the two signals. More details on the

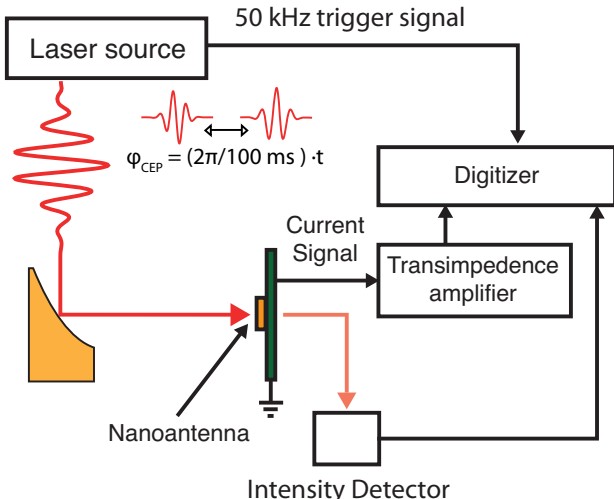

**Fig. 3 | Supplementary Fig. - Experimental setup.** The experimental setup consists of a home-built laser source, delivering 18 fs pulses at a center wavelength of 2.7 µm with up to 84 nJ of energy at a repetition rate of 50 kHz, a 25.4 mm focal length off-axis parabola and the nanoantenna detector element at the focal spot. For the detection of the charge signal, we use a custom trans-impedance amplifier with a gain of 1 GV A⁻¹ and a −3 dB bandwidth of 50 kHz. For detection of the single shot intensity signal of the laser pulse, we use a 50 kHz bandwidth pyroelectric detector in transmission after the detector. The charge and the intensity signals are digitized with an 8-bit oscilloscope at a sampling rate of 20 MSa/s. To retrieve the individual single-shot events, the digitized pulses are integrated and sorted based on the timing signal of the 50 kHz trigger signal provided by the laser source. To produce a CEP-dependent signal, the CEP of the laser source is linearly swept at a rate of $2\pi/100$ ms for 600 ms.

acquisition and digitization of the signal are given in the Supplementary Information Experimental setup.

In this experiment, each dataset consisted of the measured charge from the nanoantenna array and the corresponding pulse energy, recorded for around 50, 000 shots (1 s). In each dataset, the CEP of the laser was linearly ramped for 600 ms with a speed of $20\pi$ rad s⁻¹, starting at ~120 ms. For different datasets, the pulse energy was systematically varied by more than a factor of ten.

A single dataset is presented in Fig. 4, including both the single-shot data and the moving average calculated over 150 shots (dark line). The upper panel shows the recorded charge produced by the nanoantenna array, with an average yield of 25,000 e per shot. From 120 ms to 720 ms, the CEP is linearly ramped over a $12\pi$ range. The data points show a clear sinusoidal CEP dependence with an amplitude of 2370 e and a signal-to-noise ratio (SNR) of 4.6, while the pulse energy does not show modulation. In addition, we estimated the CEP noise of our measurement to be 0.75 rad rms (see Complementary measurements in Supplementary Information for the details of the estimation). When considering the number of illuminated antennas, the individual CEP-sensitive yield per antenna and shot is 3.3 e, we estimated peak currents through the nanoantenna gap of up to 95 e/fs, corresponding to ~15 mA. Given the surface area of a single nanoantenna tip, ~628 nm², the estimated current density reaches a remarkable 2.4 GA cm⁻². At $t = 370$ ms and 620 ms, sharp changes are visible in the charge yield of the detector element. These features, which are 250 ms apart, are caused by the specific movement pattern of the closed-loop slip-stick piezo stage used to control the CEP, which recenters the piezo position every 1.3 µm.

To isolate the CEP-dependent signal from readout noise and pulse energy fluctuations, we Fourier transformed the dataset between $t = 120$ ms and $t = 620$ ms and compared it to the frequency spectrum obtained without any optical input; see Fig. 5.

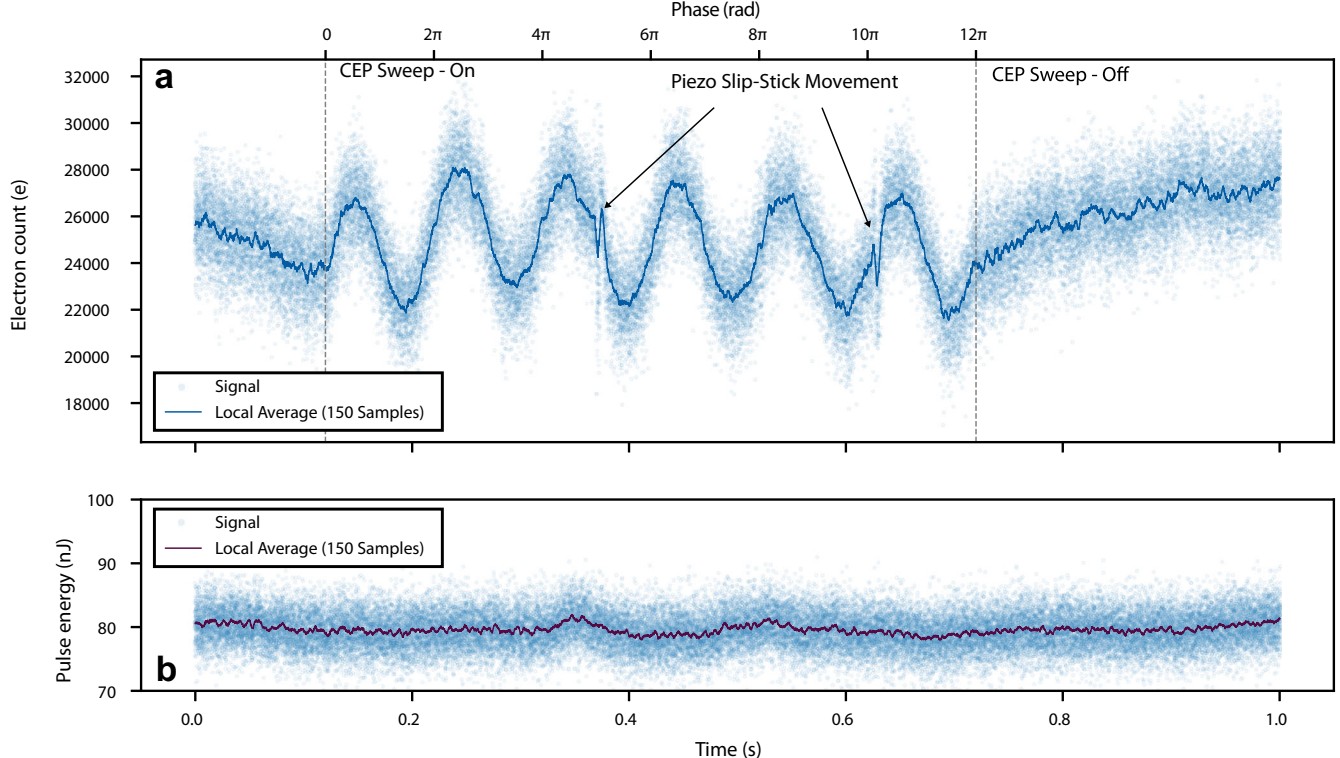

**Fig. 4 | Single-shot charge readout.** single dataset recording of 50 000 laser shots for the charge yield of the nanoantenna detector (upper panel **a**) and the laser energy recorded by the pyroelectric detector (bottom panel **b**). The peak field of the incident laser pulse on the array is 1.6 V nm. From 120 ms to 720 ms, the CEP was linearly ramped over 6 cycles. The instantaneous phase was interpolated with the scan speed of $2\pi s^{-1}$.

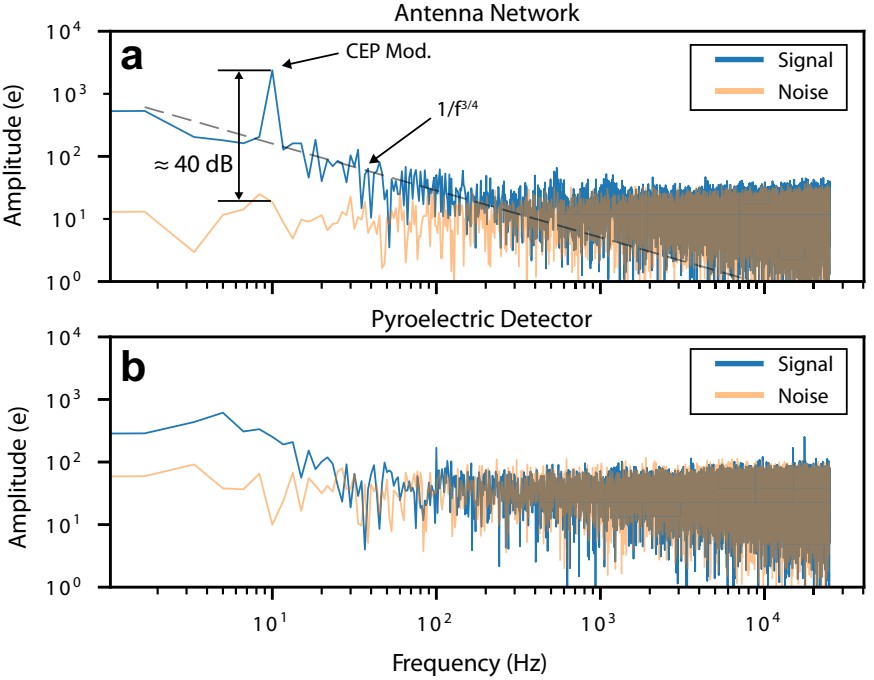

**Fig. 5 | Frequency Domain of the single-shot data.** The respective data from Fig. 4 $t = 370$ ms to $t = 620$ ms is Fourier transformed and shown in charge amplitude as a function of frequency. For comparison, the electronic noise floor is shown in orange for both spectra. **a** The frequency-resolved signal of the nanoantenna network. The 10 Hz CEP modulation is separated by 40 dB from the noise floor. **b** The frequency-resolved pulse energy fluctuation, detected with the pyroelectric detector.

The spectrum of the antenna array shows a clear peak at 10 Hz corresponding to the $2\pi \cdot 10$ Hz modulation of the CEP. This signal amplitude is around two orders of magnitude (40 dB) higher than the readout noise floor. The noise in the measured spectrum is dominated from DC to ~250 Hz by $f^{-3/4}$ scaling, which is typical for field emission devices and is attributed to Brownian noise of the work function due to dynamical changes of adsorbates on the surface[18,34]. At frequencies higher than 250 Hz, the spectrum is limited by shot noise, with a substantial component originating from the detection noise of the trans-impedance amplifier. The calculated shot noise of the signal is ~160 e rms. We also want to note that we did not observe noticeable degradation of the devices in comparison to studies using oscillator-type laser sources with MHz-level repetition rates, where degradation was present on the few-minute time scale[18]. However, detailed studies of durability and lifetime are certainly warranted, as has been carried out at DC field emission with comparable devices over 2500 h[35]. When evaluating the recorded pulse energy fluctuations at the photo-detector, no 10 Hz modulation is distinguishable from the background (see Fig. 5b). Above 100 Hz, the pulse energy spectrum is dominated by detector noise. Systematic investigation of signal strength as a function of peak electric field has shown that at ~1 Hz resolution bandwidth, a signal distinguishable from noise can be observed down to 0.6 V nm$^{-1}$ (corresponding to ~10 nJ). See Complementary measurements in Supplementary Information for details.

## Discussion

We have demonstrated a single-shot readout of CEP-dependent charge signals at a 50 kHz repetition rate, underlying sub-cycle current generation across a macroscopic device area of 225 µm$^2$ integrating more than 700 individual antenna pairs. This was made possible by improving the average CEP-dependent charge yield per single antenna by a factor of ~30[18,21], now reaching 3.3 e per shot, and by illuminating hundreds of antennas simultaneously. The enhanced antenna yield implies a remarkable peak current density estimated to reach up to 2.4 GA cm$^{-2}$[9,18,21]. With this result, we show that metallic nanoantenna networks, fabricated via state-of-the-art lithographic methods, are a flexible and scalable approach to optical-frequency electronics that allows the designing of individual circuit elements, similar to conventional microelectronics. Thanks to this advance, we demonstrated off-resonant antennas that are sensitive to pulse energies two orders of magnitude lower than any other photoemission-based single-shot absolute CEP detection techniques[15,24,36] and comparable to or lower than f-2f interferometry[37], enabling absolute CEP detection of optical pulses with only tens of nanojoules of energy. Further optimization of the network density (see Electromagnetic simulation of the nanoantenna in Supplementary Information) combined with a reduced number of optical cycles in the pulse would potentially increase the total yield by an additional two orders of magnitude[18,33]. As the measurement is dominated by read-out noise, further noise reduction of electronics downstream of the detector element will have a significant impact on SNR with the potential for another 5- to 10-fold improvement[38]. It is generally expected that when using optimized detector circuits, the SNR will also be dictated by the performance of the trans-impedance amplifier. Transimpedance amplifiers generally provide less noise and higher gain at lower bandwidth operation, typically resulting in higher SNR when using lower repetition rate laser sources. In contrast, when using higher repetition rates laser sources, the trans-impedance amplifier will exhibit higher noise levels and reduced gain, reducing the SNR. In addition, very high repetition rates may also lead to increased heat load on the devices requiring a reduced pulse energy to avoid damage which limits the overall single-shot SNR performance. With these improvements and technical optimizations the measured phase noise of 0.75 rad will be lowered down to tens of milliradians and soon competitive with established techniques, but integrated fully on a chip and with a compact detector footprint.

Given the exceptional current densities generated in these nanometer-sized devices, further studies will be necessary to elucidate the role of electron-electron interaction during the sub-cycle emission process[39]. Based on this platform, many different experiments and applications can be developed either based on single nanoantennas or larger network structures. Specifically, single antennas are interesting for the investigation of petahertz-bandwidth logic gates and memory cells[16,17]. For network structures, the small device size, comparable to the pixel size in modern Si-based CMOS detectors, combined with the reduced pulse energy requirements, enables the integration of multiple nanoantenna arrays in a larger pixel matrix. This will allow for a CEP-sensitive camera with further improved noise performance[40]. Absolute single-shot CEP tagging can also be implemented by adapting I/Q detection with two separate networks recording $\pi/2$ phase-shifted currents. The previously demonstrated techniques of attosecond-resolved field sampling can be extended to single-shot readout, by making large line arrays of individual networks[12,41]. Another area of progress will be the adaptation of the fabrication process to become fully CMOS-compatible by replacing gold with aluminum or copper.

More broadly, pushing the boundaries in petahertz electronics will require future investigations of new device classes such as transistors and logic circuits and also new material platforms. With our results we illustrate a path towards scalable and directly applicable petahertz electronics.

## Methods
### Laser source

The two-cycle MIR source used to illuminate the nanoantenna networks is a home-built system based on adiabatic difference frequency generation (DFG)[42] and details can be found in ref. 23. The setup is based on a commercial Yb:KYW regenerative amplifier with a center wavelength of 1.03 µm a pulse duration of 425 fs, delivering up to 120 µJ at a repetition rate of 50 kHz. The first stage of the optical setup consists of a non-collinear optical parametric amplifier seeded with white light and pumped by the second harmonic of the pump laser[43]. The amplified seed has an energy of approximately 1.8 µJ at a center wavelength of 740 nm. After the amplification, the seed is stretched for pre-compensation of the later acquired MIR dispersion, and the pulse energy is controlled by an anti-reflection-coated metallic neutral-density filter wheel. For generation of the passively CEP stable DFG output in the MIR, the amplified seed and the stretched pump laser of ~10 ps propagate collinear through an adiabatically poled Mg:LiNbO$_3$ crystal with an identical design to Krogen et al.[42]. The generated broadband MIR pulse covers the spectral range of 2 µm to 4.5 µm at an energy of up to 84 nJ and is compressed through dispersion in BaF$_2$ and silicon. The generated pulse has a duration down to 16 fs (FWHM) at a center wavelength of 2.7 µm, characterized by a two-dimensional spectral shearing interferometry setup. The passive CEP stability of the MIR pulse inherent to the difference frequency generation process is measured with an f-2f interferometer to 190 mrad rms over 15 min.

### Nanofabrication

A fused silica wafer was purchased from MTI Corporation and cut with a die saw. The substrates were cleaned by sonicating in acetone and isopropyl alcohol for 5 min each. Subsequently, the pieces were cleaned using an oxygen plasma. Poly(methyl methacrylate) A2 was spun at 2500 revolutions per minute and baked at 180 °C, then DisCharge H2O (DisChem Inc.) was spun at 1000 revolutions per minute so that charging did not occur during the electron beam lithography write.

Electron beam lithography was performed using an electron-beam energy of 125 keV with doses varying from 4000–6000 µC cm$^{-2}$ with an applied proximity effect correction. After exposure, the resist was developed in a 3:1 isopropyl alcohol/methyl isobutyl ketone solution for 50 s at 0 °C. Subsequently, the antenna deposition was performed using an electron beam evaporator operating below $9 \cdot 10^{-7}$

Torr. First, a 2 nm adhesion layer was deposited, then 20 nm of gold. Lift-off was performed in a 65 °C–70 °C bath of N-methylpyrrolidone.

After antenna fabrication, contacts were patterned by photolithography using a bilayer of PMGI and S1838, both spun at 4500 revolutions per minute. The deposition was performed by electron beam evaporation with a 40 nm adhesion layer and 160 nm of gold so that they could be wire-bonded to a printed circuit board.

## Data availability
The measurement data generated in this study have been deposited in the figshare database under accession code 10.6084/m9.figshare.25114634.v2 Link.

## Code availability
The code to process the data and prepare the data plots of the main text and the supplementary information is available in the form of a Python Jupyter notebook at GitHub.

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

## Acknowledgements
The authors, F. Ritzkowsky and P.D. Keathley thank William Putnam for many insightful scientific discussions on the experiments. F.Ritzkowsky thanks Maximilian Kubullek for helpful discussions on CEP detection. H. Cankaya and F. Ritzkowsky thank Haim Suchowski for designing the

ADFG crystal used for this work. F. Ritzkowsky and M. Yeung thank Marco Turchetti for their contributions to the sample fabrication process. F. Ritzkowsky thanks Guido Meier for fruitful discussions on the current detection schemes. M. Budden and T. Gebert thank Johannes Kunsch and Laser Components Germany GmbH for their support in the development of pyroelectric sensor elements specifically tailored to this high-speed application. F. Ritzkowsky and all authors affiliated with Deutsches Elektronen-Synchrotron (DESY) thank the administrative and engineering support staff at DESY for assisting with this research. We thank John Simonaitis and Stewart Koppell for their assistance in reviewing the manuscript. F.X. Kärtner and F. Ritzkowsky acknowledge funding by: European Union's Seventh Framework program 145 (FP7/2007-2013) ERC Synergy Grant 'Frontiers in Attosecond X-ray Science: Imaging and Spectroscopy' (AXSIS) (609920); Cluster of Excellence 'CUI: Advanced Imaging of Matter' of the Deutsche Forschungsgemeinschaft (DFG)—EXC 2056—project ID 390715994; Deutsche Forschungs Gemeinschaft (DFG)— project ID KA908/12-1 and project ID 453615464. Felix Ritzkowsky acknowledges the Alexander-von-Humboldt Foundation for support during the writing and data analysis stage of this manuscript. We acknowledge the use of a PIER Hamburg grant for supporting M. Yeung to travel to DESY to set up initial measurements. The initial stages of this research, in particular, device design, fabrication, and experimental testing, were supported by the Air Force Office of Scientific Research (AFOSR) grant under contract NO. FA9550-18-1-0436. Later stages of this research, namely data analysis and manuscript developments, were supported by the National Science Foundation under Grant No. 2238575. M. Yeung acknowledges support from the National Science Foundation Graduate Research Fellowship Program, Grant No. 1745302. This work was carried out in part through the use of MIT.nano.

## Author contributions

F.R., M.Y., F.K., and P.K. conceived the experiments. The samples were fabricated by M.Y. with guidance from P.K. and K.B. The experimental setup was constructed by F.R. with assistance from E.B., G.R., R.M., and H.C. The readout electronics were set up by F.R., T.G., M.B., and T.M. The data was taken by F.R. The data was analyzed by F.R. with input from M.Y. and P.K. The electromagnetic simulations were done by E.B. with input from F.R., M.Y., and P.K. The manuscript was written by F.R. with significant contributions from M.Y., P.K., and G.R. and editing from all authors.

## Funding

## Competing interests

The authors declare no competing interests.
