## [Transparent Peer Review file · Nature Communications]

On-Chip Petahertz Electronics for Single-Shot Phase Detection

Corresponding Author: Mr Felix Ritzkowsky

This manuscript has been previously reviewed at another journal. This document only contains reviewer comments, rebuttal and decision letters for versions considered at Nature Communications.

Version 0:

Reviewer comments:

Reviewer #1

(Remarks to the Author)

I have reviewed the manuscript before (reviewer 1). I thank the authors for their explanations. I agree that the proof of concept of the device in addition to its increased capabilities when compared to e.g. a Stereo-ATI CEP phase meter, are significant advances. I support publication of the manuscript in its current form in Nature Communications.

Reviewer #2

(Remarks to the Author)

This is beautiful work. Single-shot detection of CEP changes with a room-temperature device at the full repetition rate of a femtosecond oscillator is an impressive achievement. Moreover, the authors have further improved the discussion. I am willing to support the publication if the authors improve the following two remaining points:

1. They have made a sincere effort to clarify that the peak currents they estimate result from a theoretical model. In this context, there is basically only one ambiguous statement left, in which the manuscript pretends to know how the evolution of electron currents occurs on the sub-cycle scale. On page 5, the text reads: "Additionally, the tunnel ionization is temporally confined to the peak regions of the strongest half-cycles of the exciting field[4, 8, 9, 24, 25, 30–32]." As explained in previous reviewer comments, the literature has shown that these dynamics can occur, but there is no guarantee that this is actually the case for all randomly selected antennas. It is an assumption that the authors make here. It should therefore be clearly labeled as such. I consider this important enough to make my approval conditional on this correction.

2. On a slightly lighter note: While the authors are correct in their assertion that the term "array" may understate the fact that all the antennas are electrically contacted, I think the term "network" is also misleading: the authors emphasize the attractiveness of their antennas to achieve PHz bandwidths, but the antennas are only electrically connected in terms of their DC (or narrowband) conductivity, not in terms of PHz response. I suggest introducing the device at first mention as an "interconnected antenna array". Once the difference between DC and PHz coupling is clarified, I can live with calling the sample a "network".

Manuscript Changes

We would like to thank the reviewers for taking the time to review our manuscript and appreciate all the feedback we received.

In the following we have responded to the individual reviewer comments and questions. Our response is highlighted in **blue** and changes to the manuscript are highlighted in **orange**. The corresponding changes in the manuscript itself are also highlighted in **orange**. The responses are sorted for the specific round of reviews with the latest review round first.

Round 3

Reviewer 1:

"I have reviewed the manuscript before (reviewer 1). I thank the authors for their explanations. I agree that the proof of concept of the device in addition to its increased capabilities when compared to e.g. a Stereo-ATI CEP phase meter, are significant advances. I support publication of the manuscript in its current form in Nature Communications."

We thank the reviewer for his/her support of publication and the previous reviews which undoubtedly strengthened this manuscript.

Reviewer 2:

"This is beautiful work. Single-shot detection of CEP changes with a room-temperature device at the full repetition rate of a femtosecond oscillator is an impressive achievement. Moreover, the authors have further improved the discussion. I am willing to support the publication if the authors improve the following two remaining points:

1. They have made a sincere effort to clarify that the peak currents they estimate result from a theoretical model. In this context, there is basically only one ambiguous statement left, in which the manuscript pretends to know how the evolution of electron currents occurs on the sub-cycle scale. On page 5, the text reads: "Additionally, the tunnel ionization is temporally confined to the peak regions of the strongest half-cycles of the exciting field[4, 8, 9, 24, 25, 30–32]." As explained in previous reviewer comments, the literature has shown that these dynamics can occur, but there is no guarantee that this is actually the case for all randomly selected antennas. It is an assumption that the authors make here. It should therefore be clearly labeled as such. I consider this important enough to make my approval conditional on this correction.
2. On a slightly lighter note: While the authors are correct in their assertion that the term "array" may understate the fact that all the antennas are electrically contacted, I think the term "network" is also misleading: the authors emphasize the attractiveness of their antennas to achieve PHz bandwidths, but the antennas are only electrically connected in terms of their DC (or narrowband) conductivity, not in terms of PHz response. I suggest introducing the device at first mention as an "interconnected antenna array". Once the difference between DC and PHz coupling is clarified, I can live with calling the sample a "network".

We thank the reviewer for his/her support of publication and gladly incorporate his/her remaining feedback to further improve our manuscript.

On point 1, we agree with the reviewer and thank him/her for pointing out the possible ambiguity. We modified the sentence on page 5 into the following,

"Additionally, theoretical models predict that the tunnel ionization is temporally confined to the peak regions of the strongest half-cycles of the exciting field [4, 8, 9, 24, 25, 30–32]."

On point 2, we also agree with the reviewer and did not intend to suggest that the whole array responds on the PHz bandwidth, but rather as individual PHz devices embedded in a lower bandwidth network of many parallel devices collectively forming the signal that is ultimately collected with a kHz bandwidth.

We clarified this notion on page 3,

"To circumvent damage the pulse energy can be distributed over a network of nanoantennas, which respond individually at a PHz bandwidth to the optical-field, but collectively contribute their produced charge signal to the network, which is subsequently read out at radiofrequencies."

Other Changes

We updated the acknowledgments to better reflect the funding that supported the late stages of writing and incorporating feedback by the review process.